# Gauging the variational optimization of projected entangled-pair states

**Wei Tang[1] [⋆], Laurens Vanderstraeten[2] and Jutho Haegeman[1]**

**1** Department of Physics and Astronomy, Ghent University, Krijgslaan 281, 9000 Gent, Belgium
**2** Center for Nonlinear Phenomena and Complex Systems, Université Libre de Bruxelles, CP 231, Campus Plaine, B-1050 Brussels, Belgium

⋆ wei.tang.phys@gmail.com

## Abstract

Projected entangled-pair states (PEPS) constitute a powerful variational ansatz for capturing ground state physics of two-dimensional quantum systems. However, accurately computing and minimizing the energy expectation value remains challenging, in part because the impact of the gauge degrees of freedom that are present in the tensor network representation is poorly understood. We analyze the role of gauge transformations for the case of a U(1)-symmetric PEPS with point group symmetry, thereby reducing the gauge degrees of freedom to a single class. We show how gradient-based optimization strategies exploit the gauge freedom, causing the tensor network contraction to become increasingly inaccurate and to produce artificially low variational energies. Furthermore, we develop a gauge-fixed optimization strategy that largely suppresses this effect, resulting in a more robust optimization. Our study underscores the need for gauge-aware optimization strategies to guarantee reliability of variational PEPS in general settings.

# 1   Introduction

Tensor network states [1, 2] have become a powerful and widely used variational ansatz for low-entanglement eigenstates of local Hamiltonians. The class of projected entangled-pair states (PEPS) [3], a natural extension of matrix product states (MPS) to two dimensions, has proven powerful but remains significantly more challenging to handle numerically. In contrast to MPS, PEPS expectation values cannot be evaluated exactly, but have to be computed through approximate contraction schemes such as the corner transfer matrix renormalization group [4–7] or boundary MPS methods [8–12]. The optimization of PEPS ground-state approximations was initially done using imaginary-time evolution with various truncation schemes, giving rise to techniques such as the simple update [13], the full update [8], and the fast full update [14]. More recently, gradient-based optimization strategies were found to yield better variational energies. Initially, energy gradients were constructed manually [15, 16], but the application of automatic differentiation (AD) [17] has since made this approach far more practical and accessible [18–21]. As a result, variational infinite PEPS calculations are now among the standard tools to simulate strongly interacting quantum lattice models in two dimensions [21–27].

The use of AD, however, also introduces a new concern: AD computes the gradient of the energy evaluated through approximate tensor network contractions. The optimization may thus exploit errors in the approximations, yielding artificially low energies rather than genuine improvements to the physical state. An additional concern is the presence of gauge degrees of freedom in the PEPS representation. MPS algorithms typically exploit the gauge freedom to construct canonical forms that enhance their stability; for PEPS, a canonical form with similar favorable properties does not exist in general. There have been proposals for fixing the gauge degrees of freedom in PEPS [28,29] or for approximate canonical forms [30], and symmetry constraints can be imposed on the PEPS tensors that rule out additional gauge transformations [12,31,32]. In the general case, however, the impact of the gauge degrees of freedom on PEPS algorithms remains unclear.

Both concerns are tightly coupled, as it was recently shown that gauge transformations can drastically affect the precision of contracting two-dimensional tensor networks [33]. This

originates from the fact that approximate tensor network contraction schemes typically utilize the entanglement structure of the environment, which can be modified by non-unitary gauge transformations. Put differently, the environment does not transform covariantly under a non-unitary gauge transform on the virtual bonds of the PEPS, making the approximate energy (and other expectation values) depend on the PEPS gauge. This observation raises serious questions on how virtual gauge transformations impact the variational optimization of PEPS in practice, and to what extent gauge fixing can improve its performance.

In this work, we investigate these questions for a restricted PEPS ansatz that combines internal U(1) and point group symmetries on the square lattice. This construction leaves only a single class of gauge transformations, allowing us to easily analyze gauge dependence and to develop an optimization algorithm on the manifold of gauge-fixed PEPS. In particular, we directly compare the performance of unconstrained and gauge-fixed PEPS optimizations.

## 2 The PEPS ansatz

We will consider the protoypical Bose-Hubbard model [34] on the square lattice, described by the hamiltonian

$$H = -\sum_{\langle i,j \rangle} \left( a_i^\dagger a_j + \text{h.c.} \right) + \frac{U}{2} \sum_i n_i (n_i - 1), \tag{1}$$

where $a_i^\dagger$ and $a_i$ are the creation and annihilation operators, respectively, and $n_i = a_i^\dagger a_i$ is the number operator. We will work at integer fillings, for which a zero-temperature transition between a U(1) conserving Mott-insulating phase and a symmetry-broken superfluid phase occurs; for unit filling, this phase transitions occurs around $U \approx 16.7$ [35, 36].

For describing the ground state in the Mott-insulating phase, we introduce an infinite PEPS ansatz that combines internal and spatial symmetries. The ansatz is composed of local site tensors $A$ and bond tensors $D$, located on the vertices and edges of the square lattice, respectively. The tensors $A$ and $D$ can be considered as linear maps between U(1)-graded vector spaces, with their directions indicated by arrows in Fig. 1. Tensor $A$ is defined as a linear map from $\mathcal{V}^{\otimes 4}$ to $\mathcal{P} \otimes \mathcal{P}_a$, and tensor $D$ is defined as a linear map from an empty space to $\mathcal{V}^{\otimes 2}$. Here, $\mathcal{V}$ is the virtual space of the PEPS, $\mathcal{P}$ is the local bosonic Hilbert space, and $\mathcal{P}_a$ is an auxiliary space of dimension 1. By assigning a non-zero U(1)-charge $q_a$ to the auxiliary space $\mathcal{P}_a$, we can fix the physical particle density of the PEPS to be $\rho = -q_a$.

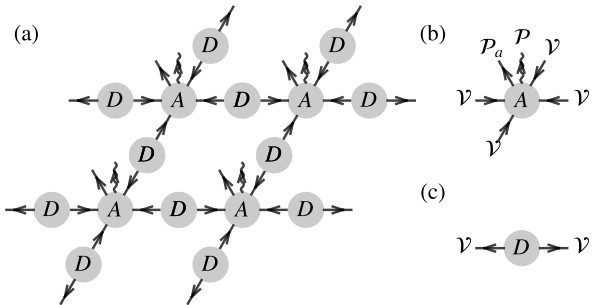

Figure 1: (a) The U(1)-symmetric PEPS ansatz. The wavy lines represent the physical space, the open solid lines represent the auxiliary space, and the contracted solid lines represent the virtual space. (b) The site tensor $A$ located on the lattice sites. (c) The bond tensor $D$ located on the edges of the square lattice.

We consider the case where the ground state is invariant under rotations and reflections, allowing us to impose $C_{4v}$ symmetry on the local PEPS tensors [31, 32, 37, 38]. This effectively

fixes most of the gauge degrees of freedom on the virtual indices without significantly reducing the expressive power of the PEPS ansatz. To achieve this, we first note that the tensor $D$ can be fixed as a constant tensor, with its non-zero blocks set to identity matrices. This ensures that $D$ is invariant under the $C_{2v}$ point group operations. Next, we require the site tensor $A$ to transform according to a specific irreducible representation of the $C_{4v}$ point group. An orthonormal basis $\{A_j\}$ can be constructed such that each $A_j$ satisfies both the U(1) symmetry and the point group symmetry. Any tensor $A$ can then be expressed as a linear combination of these basis tensors, $A = \sum_j \alpha_j A_j$, where $\alpha_j$ are scalar coefficients [see Appendix A for more details]. In this work, we focus on the $A_1$ representation of $C_{4v}$ and restrict all coefficients $\alpha_j$ to be real, which ensures that the resulting tensors are real.

## 3 Virtual gauge transformations

We consider gauge transformations that preserve the restricted form of our PEPS ansatz. Hereto, we transform the local site tensor as

$$(2)$$

where $G$ is a U(1)-invariant matrix that furthermore satisfies

$$(3)$$

so that the bond tensor is left constant. Because unitary gauge transformations do not contribute to the problems we seek to investigate, we can restrict to $G$ being positive definite [see Appendix B for more details], which we parameterize as $G = \exp(\tau N)$ with $N$ a symmetric, U(1)-invariant matrix, that furthermore satisfies the linear condition $ND + DN = 0$ because of Eq. (3). Such matrices form a vector space, so that we can find a basis $\{N_\gamma\}$ with respect to which we can expand arbitrary $N$.

If we then consider an infinitesimal transformation $G = \exp(\tau N)$ for $\tau \to 0$, the transformation of the PEPS tensor can be expressed as $A \mapsto A + \tau X$ with $X$ given by

$$(4)$$

Since the mapping from $A$ to $X$ is a linear transformation, for each choice of $N$, there exists a symmetric matrix $\sigma$ so that $X = \sum_{jk} \sigma_{jk} \alpha_k A_j$ for $A = \sum_k \alpha_k A_k$ [see Appendix C for more details]. Each basis element $N_\gamma$ thus has an associated matrix $\sigma_\gamma$ that representations the action of an infinitesimal transformation with respect to the chosen tensor basis of tensors $\{A_j\}$. A gauge transformation $G = \exp(\sum_\gamma \tau_\gamma N_\gamma)$ then transforms $A = \sum_k \alpha_k A_k$ as

$$A \mapsto \sum_{jk} \left[ e^{\sum_\gamma \tau_\gamma \sigma_\gamma} \right]_{jk} \alpha_k A_j. \tag{5}$$

# 4  Gauge fixing and manifold optimization

Although the gauge degrees of freedom are identified, it is not *a priori* clear what the optimal gauge-fixing condition would be. A natural choice is the minimal canonical form (MCF) [28], which is defined as the tensor that minimizes the 2-norm of the local tensor under gauge transformations [1]. This choice is particularly appealing, because it is unique up to local unitaries [28] and it can be easily implemented numerically, especially in the current setting where only a limited number of gauge degrees of freedom remain.

For the orthonormal basis $\{A_j\}$, the MCF condition is equivalent to requiring the 2-norm of the coefficients $\sum_j |\alpha_j|^2$ to be minimized with respect to gauge transformations of the form (5), which can be further reduced to the condition

$$\vec{\alpha}^T \sigma_\gamma \vec{\alpha} = 0, \quad \forall \gamma. \tag{6}$$

In this case, it becomes straightforward to show that the solution to (6) is unique and can be obtained by minimizing the norm $\sum_j |\alpha_j|^2$ using Newton's method [see Appendix **??** for more details]. Owing to the limited number of gauge degrees of freedom, the gauge-fixing procedure introduces almost no additional computational cost.

To move on the manifold of gauge-fixed PEPS tensors, henceforth referred to as the MCF manifold, we should restrict to variations $\delta\vec{\alpha}$ in the tangent space of the manifold, which are characterized by

$$\vec{\alpha}^T \sigma_\gamma \delta\vec{\alpha} = 0, \quad \forall \gamma. \tag{7}$$

The orthogonal complement thereof is a subspace of variations of the form $\delta\vec{\alpha}_{\text{gauge}} = \sum_\gamma \mu_\gamma \sigma_\gamma \vec{\alpha}$ with arbitrary coefficients $\mu_\gamma$, which exactly corresponds to infinitesimal gauge transformations. It is a uniquely defining property of the MCF that its tangent space is exactly orthogonal (with respect to the Euclidean inner product) to the subspace of infinitesimal gauge transformations. Changing the parameters $\vec{\alpha}$ in a direction $\delta\vec{\alpha}_{\text{gauge}}$ should not affect the physical state or any physical expectation values (to first order). Therefore, if we compute the energy gradient $\Delta\vec{\alpha} = \nabla e(\vec{\alpha})$, we expect $(\delta\vec{\alpha}_{\text{gauge}})^T \Delta\vec{\alpha} = 0$ for all $\delta\vec{\alpha}_{\text{gauge}}$, or thus for the gradient $\Delta\vec{\alpha}$ to live in the tangent space of the MCF manifold. However, it turns out that in practical calculations, this condition is not satisfied because the energy and its gradient does contain gauge-dependent contributions resulting from the aforementioned lack of gauge covariance in the environment calculation.

If we restrict the optimization to gauge-fixed PEPS, we can project the gradient onto the tangent space of the MCF manifold. This is equivalent to redefining $\Delta\vec{\alpha} \mapsto \Delta\vec{\alpha} + \sum_\gamma \mu_\gamma \sigma_\gamma \vec{\alpha}$ with coefficients $\mu_\gamma$ determined by the linear system

$$\vec{\alpha}^T \sigma_\gamma \left( \Delta\vec{\alpha} + \sum_{\gamma'} \mu_{\gamma'} \sigma_{\gamma'} \vec{\alpha} \right) = 0, \quad \forall \gamma. \tag{8}$$

It is worth noting that the solution to (8) also minimizes the Euclidean 2-norm $\|\Delta\vec{\alpha} + \sum_\gamma \mu_\gamma \sigma_\gamma \vec{\alpha}\|$ of the new gradient, which is again a property of the MCF and naturally aligns with the overall goal of optimization—reducing the norm of the gradient.

After determining a search direction $\delta\vec{\alpha}$, the linear retraction $\vec{\alpha} \to \vec{\alpha} + s\delta\vec{\alpha}$ introduces a deviation from the MCF manifold of order $O(s^2)$. To ensure that the PEPS tensor remains on the manifold, the linear retraction can be complemented with an additional gauge fixing step, or more sophisticated retraction schemes can be devised. These ingredients enable to

---

[1]One caveat is that, if one releases the spatial symmetry constraint, the minimal canonical form may not give rise to a tensor that satisfies the spatial symmetry locally. Here, we abuse the term "minimal canonical form" to refer to the tensor that minimizes the 2-norm of the local tensor under gauge transformations of the form (2).

formulate a gradient descent optimization on the MCF manifold. To perform more efficient quasi-Newton methods such as L-BFGS, however, one should also construct a vector transport, i.e., appropriately transporting tangent vectors from previous iterations into the tangent space of the current iteration [39]. Details of the retraction and the vector transport are provided in Appendix E.

## 5 Optimization results

We can now compare the performance of the gauge-fixed and unconstrained PEPS optimizations for the example of the Bose-Hubbard model [Eq. (1)] at $U = 30$ and unit fulling $\rho = -q_a = 1$, placing the system deep in the Mott insulating phase with unbroken U(1) symmetry. The virtual space of the PEPS is chosen to be $\mathcal{V}_0^{\oplus 2} \oplus \mathcal{V}_1^{\oplus 1} \oplus \mathcal{V}_{-1}^{\oplus 1}$, where $\mathcal{V}_q$ denotes a one-dimensional vector space carrying the irreducible representation of U(1) with charge $q$, yielding a total PEPS bond dimension of 4. For this choice of virtual space, there is only one (non-unitary) gauge degree of freedom left, i.e., the space of (symmetric) $N$ matrices that anti-commute with the bond tensor $D$ is one-dimensional and spanned by the matrix

$$N_{q=1} = 1, \quad N_{q=-1} = -1, \quad N_{q=0} = \begin{pmatrix} 0 & 0 \\ 0 & 0 \end{pmatrix}. \tag{9}$$

Since the system is deep in the Mott phase, where the ground state is close to a product state, we furthermore utilize the perturbative construction [40] to obtain a good initial PEPS. We then perform gradient-based optimization of the PEPS, using the L-BFGS algorithm. The tensor network contraction is carried out using the boundary MPS method, with the fixed-point MPS obtained via the standard variational uniform MPS (VUMPS) algorithm [41, 42]. For all data presented, the VUMPS algorithm converges with a convergence measure of $10^{-9}$. The gradient of the energy with respect to the PEPS tensors is computed with AD [17, 21, 43]. Throughout the optimization, the virtual space of the boundary MPS is fixed to $\mathcal{V}_0^{\oplus 1} \oplus \mathcal{V}_1^{\oplus 2} \oplus \mathcal{V}_{-1}^{\oplus 2}$, resulting in a total environment bond dimension of $\chi = 5$. Details of the perturbative construction and the boundary MPS calculation are provided in Appendix F.

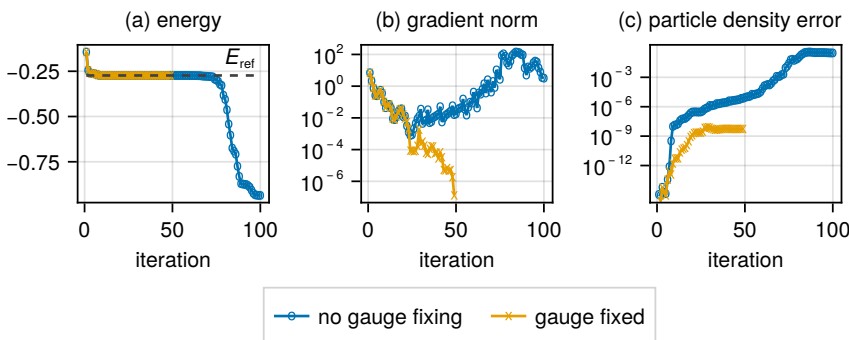

Figure 2: Optimization results for the PEPS optimization with the gauge-fixed PEPS tensors and unconstrained PEPS tensors, respectively, with environment bond dimension $\chi = 5$. We show (a) the measured energy, (b) the gradient norm, and (c) the error in the measured particle density. The dashed line in subplot (a) is a reference energy $E_{\text{ref}}$.

The results of the optimization are shown in Fig. 2. The energies are compared to a reference energy $E_{\text{ref}}$, obtained from the gauge-fixed PEPS optimization with an environment bond

dimension $\chi = 15$, details of which are given below. For the unconstrained PEPS optimization, the measured energy appears unbounded from below, and the gradient norm initially decreases but subsequently increases, consistent with the behavior of the measured energy. These low "energies" are clearly not reliable, as confirmed by measuring the particle density with the same environment, which deviate from the exact value of 1. In contrast, the gauge-fixed PEPS optimization quickly converges, with the measured particle density remaining very close to the exact value of 1 throughout the optimization.

One might find the unbounded energy behavior in the unconstrained PEPS optimization unsurprising, since the tensor network contraction is only performed approximately with a very small environment bond dimension $\chi = 5$, and the variational principle is only recovered in the limit $\chi \to \infty$. However, we clearly see that the gauge-fixed PEPS optimization with the same $\chi$ still converges to a reasonable ground-state energy, indicating the existence of a local minimum on the gauge-fixed PEPS manifold close to the true optimal $D = 4$ PEPS state. This local minimum energy may still lie below the actual ground-state energy of the system, but this does not pose a serious problem for the optimization, as energy improvements can be achieved by increasing the environment bond dimension further. Along this line, it becomes clear that the loss of the variational principle does not necessarily cause energy instability, and the instability observed in the unconstrained PEPS optimization mainly arises from exploiting the unfixed virtual gauge degrees of freedom.

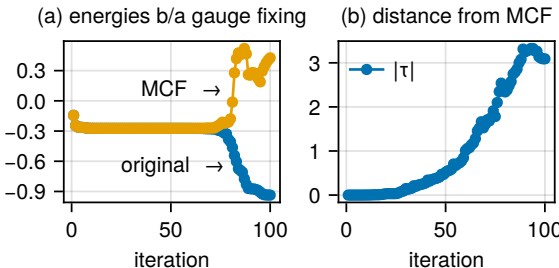

Figure 3: Additional analysis of the PEPS tensors from the unconstrained PEPS optimization. (a) The energies measured after bringing the unconstrained PEPS tensors into the MCF, compared to the energies measured with the original unconstrained PEPS tensors. (b) The distance of these unconstrained PEPS tensors from their MCF, represented by $|\tau|$ in $G = \exp(\tau N)$ [c.f. Eq. (2)].

To further confirm the above analysis, we take the PEPS tensors from each iteration of the unconstrained PEPS optimization, fix them to the MCF gauge, and recompute the boundary MPS to obtain the corresponding energies. The energies measured with the MCF-gauge-fixed PEPS tensors are shown in Fig. 3(a), alongside the energies obtained from the original unconstrained PEPS tensors. It becomes clear that the unconstrained PEPS tensors yielding artificially low energies actually correspond to high-energy PEPS states, and that the unconstrained optimization is driving the PEPS away from the true ground state. The unconstrained PEPS optimization reaches these high-energy states by exploiting the virtual gauge degrees of freedom and introducing large errors in the energy measurements. This is further supported by Fig. 3(b), which shows that the strength of the gauge transformation required to bring these PEPS tensors into the MCF increases as the optimization progresses.

# 6 Effects of larger environment bond dimensions

Next, we discuss how the effect of virtual gauge degrees of freedom changes with the environment bond dimension. We set the virtual space of the boundary MPS to be (i) $\mathcal{V}_0^{\oplus 2} \oplus \mathcal{V}_1^{\oplus 3} \oplus \mathcal{V}_{-1}^{\oplus 3} \oplus \mathcal{V}_2^{\oplus 1} \oplus \mathcal{V}_{-2}^{\oplus 1}$ (total bond dimension $\chi = 10$), and (ii) $\mathcal{V}_0^{\oplus 3} \oplus \mathcal{V}_1^{\oplus 5} \oplus \mathcal{V}_{-1}^{\oplus 5} \oplus \mathcal{V}_2^{\oplus 1} \oplus \mathcal{V}_{-2}^{\oplus 1}$ (total bond dimension $\chi = 15$). Using the PEPS tensor from step 20 of the unconstrained PEPS optimization at $\chi = 5$ as starting point, we perform the both unconstrained and gauge-fixed PEPS optimizations for each choice of the virtual spaces above.

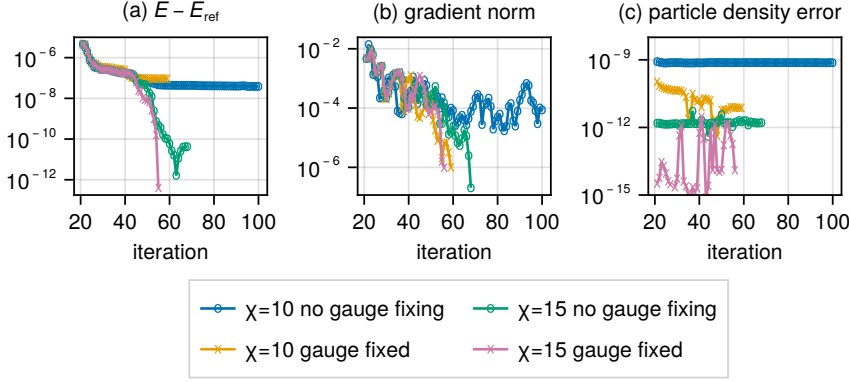

Figure 4: Results of unconstrained PEPS optimization with larger environment bond dimensions. We show (a) the measured energies compared to $E_{\mathrm{ref}}$, (b) the gradient norm, and (c) the error in the measured particle density.

From Fig. 4, we observe that the instability in the unconstrained PEPS optimization is reduced as the environment bond dimension increases. This behavior can be understood as follows. The gradient in the unconstrained optimization always contains two components: one corresponding to genuine physical variations and a component that arises from gauge dependence in the cost function, *i.e.* the approximate energy evaluation, through the environment calculation. Because the exact energy is gauge independent, we can fully attribute the second contribution to the error between the approximate and the exact energy. As $\chi$ increases, the energy error generally decreases, which in turn suppresses the gauge-dependent component of the gradient. For the cases considered, we find that unconstrained PEPS optimization only converges when $\chi = 15$, whereas it fails to converge at $\chi = 10$. Although no visible instability appears in the energy for $\chi = 10$ up to 100 iteration steps, the gradient norm fluctuates around $10^{-4}$—in sharp contrast to the behavior under gauge fixing. This suggests that, although reduced, the gauge-dependent component of the gradient continues to hinder convergence unless the environment bond dimension is sufficiently large to fully suppress the influence of virtual gauge transformations.

We note that the ground state under consideration is deep in the insulating phase and thus close to a product state, and a relatively small environment bond dimension of $\chi = 15$ is sufficient to fully suppress the effects of virtual gauge degrees of freedom. In more challenging problems, where the ground state typically exhibits significant entanglement, much larger—possibly infeasible—environment bond dimensions may be required to fully suppress the influence of gauge transformations. However, our results illustrate that it can be possible to obtain a correctly optimized energy and state with significantly smaller values of $\chi$, provided that a gauge fixing strategy is implemented during the optimization.

## 7 Further discussions

Finally, we address the question of whether the gauge-fixed PEPS optimization on the MCF manifold is always stable. There is no *a priori* guarantee that the MCF minimizes inaccuracies in the contraction, and there may exist ill-conditioned regions within the MCF manifold where the energy measurements lead to large negative errors.

To demonstrate that such ill-conditioned regions do exist, we again consider the example of the Bose-Hubbard model. Starting from the same initial PEPS tensor, we alter the gauge-fixed PEPS optimization at $\chi = 5$ by deliberately adding a poorly constructed preconditioner [2]. The results are shown in Fig. 5. This preconditioned optimization fails to converge, with the energy function again exhibiting unbounded behavior from below. This is similar to what is observed in the unconstrained PEPS optimization, albeit the absolute value of the error remains orders of magnitude smaller than in the fully unconstrained case.

Note that the instability of the preconditioned optimization presented in Fig. 5 is not crucially dependent on the specific preconditioner being used, and is fundamentally a property of the cost function. To illustrate this, we construct a linear interpolation between the minimum found with the converged optimization and the unstable solution with lower energy obtained after 80 iteration steps of the failed optimization (which does not constitute a minimum, as indicated by the norm of the gradient). By bringing the interpolated tensor to the MCF gauge and evaluating the energy for both $\chi = 5$ and $\chi = 10$, we obtain a one-dimensional slice of the cost function. Clearly, for $\chi = 5$, an extended region of parameter space produces lower energies than the physical minimum, while this region yields significantly higher energy for the case for $\chi = 10$. It is in fact interesting to see that increasing $\chi$ (slightly) reduces the energy around the physical minimum, but increases the energy in the spurious region. This analysis also illustrates the importance of the well-constructed starting point for the earlier optimization results presented in Fig. 2. If the gauge-fixed optimization at $\chi = 5$ would have started from a random initialization, as is often the case in more challenging settings, it might well have started in the attractor basin of the unstable region. To further support the above analysis, we compare the MCF gauge with an alternative gauge-fixing condition in Appendix H.

## 8 Conclusion and outlook

Although our analysis is based on a specific example, we arrive at a general conclusion: the PEPS gradient of the approximate energy evaluation contains components along gauge directions, which are unphysical and result from the lack of gauge covariance in the construction of the environment. These components can drive the optimization into regions where tensor network contractions become ever more inaccurate, leading to artificially low energy values instead of genuinely improving the physical state. This situation is expected to become worse as less symmetries are imposed and larger unit cells are considered, so that more gauge degrees of freedom become available. In these cases, although larger environment bond dimensions $\chi$ might allow the optimization to reach a certain level of convergence, we expect that a full convergence of the PEPS optimization would require unfeasible values of $\chi$ quickly. Further-

---

[2]Roughly speaking, the preconditioner is a matrix $P$ that is applied to the gradient as $P^{-1}g$ at each optimization step, which twists the search direction of the optimization. The preconditioner that we employ is a diagonal matrix, which is defined as $P_{jj} = \sqrt{\sum_{j=1}^{k} \eta^{j-1}(g_j^{(k-j)})^2}$, where $g_j^{(k-j)}$ denotes the $j$-th component of the gradient at the $(k-j)$-th optimization step, and $\eta = 0.01$ is a small positive constant. The previous gradients are reused without any proper vector transport procedure, which leads to a poorly conditioned preconditioner. Nevertheless, since the preconditioner is positive definite, the optimization should still converge to the same minimum point if the energy function has a unique minimum.

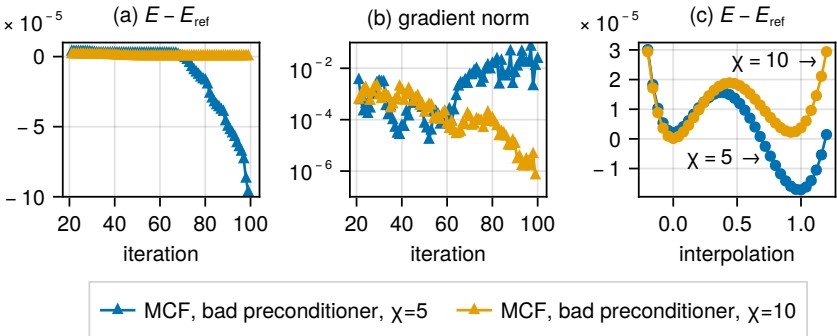

Figure 5: Gauge-fixed PEPS optimization results using the preconditioner. The optimizations are performed with environment bond dimensions $\chi = 5$ and $\chi = 10$. The $\chi = 10$ optimization is initialized from the PEPS tensor at step 20 of the $\chi = 5$ optimization. (a) The measured energies compared to $E_{\text{ref}}$. (b) The gradient norm. (c) The measured energies along a linear interpolation between two PEPS tensors: $A^{(0)}$, the converged tensor from a successful $\chi = 5$ optimization, and $A^{(1)}$, the tensor at step 80 of the failed $\chi = 5$ optimization. The energies are measured using boundary MPS with bond dimension $\chi = 5$ and $\chi = 10$. The interpolation parameters 0 and 1 correspond to $A^{(0)}$ and $A^{(1)}$, respectively.

more, obtaining well-converged physical results at smaller values of $\chi$ can be very valuable in the context of finite correlation length scaling studies [44–47].

For the case study in this paper, the optimization was stabilized by restricting the PEPS tensors to the MCF manifold and projecting the gradient accordingly. Based on our observations, we believe it will be crucial to extend this gauge-fixing approach to situations with other internal or spatial symmetries, larger unit cells and to more general PEPS without such structural constraints.

Still, even for the simple case at hand, we were able to detect regions in the MCF manifold where ill-conditioned contractions persist for low values of the environment bond dimension $\chi$. Assessing the severity of this problem for more general PEPS, and possibly investigating the existence of alternative gauge conditions that do not exhibit these pathological regions will be required. An alternative strategy might consist of devising a scheme for constructing the PEPS environment that is manifestly gauge covariant. Here, fixed point schemes based on biorthogonality might be relevant [10, 48, 49]. Yet another approach could be to revisit the analytically constructed gradient of the exact energy evaluation, which was the starting point of the earliest gradient-based PEPS studies [15, 16]. This quantity is gauge independent by construction, but cannot be evaluated exactly, and should then be approximated in a way that does not introduce gauge-dependent components. Whether such a strategy exists at all, and to what extent AD techniques can still be leveraged to facilitate the tedious parts of the computation, remains an open question.

# Acknowledgments

We acknowledge insightful discussions with Philippe Corboz, Juraj Hasik, Frank Verstraete, Lander Burgelman, Xingyu Zhang, Yuchi He, Gleb Fedorovich, Nick Bultinck, and Wei-Qiang Chen.

**Funding information**   This work was supported by the Research Foundation Flanders (FWO) [Postdoctoral Fellowship 12AA225N] and the European Union's Horizon 2020 program [Grant Agreement No. 101125822 (GaMaTeN)].

**Data availability**   The data that supports the findings of this work is openly available [50]. The code of this work is implemented with the open-source libraries `TensorKit.jl` [51] (for the internal U(1) symmetry), `SpatiallySymmetricTensors.jl` [52] (for the point group symmetry), and `OptimKit.jl` [53] (for the manifold optimization).

## A   Imposing the point group symmetry on the local PEPS tensor

The idea of imposing point spatial symmetries on PEPS tensors is systematically studied in Refs. [31, 32, 38]. In this section, we briefly review how to impose the point group symmetry on the PEPS tensor $A$, and discuss the practical procedure to construct the orthonormal basis $\{A_j\}$.

We first note that all U(1)-symmetric $A$ tensors form a linear space $\mathbb{V}_A$, and they remain within the same linear space under the action of the $C_{4v}$ group operations (such as rotations and reflections). Therefore, each element $g \in C_{4v}$ can be represented as a linear map $\rho(g) : \mathbb{V}_A \to \mathbb{V}_A$ that implements a permutation of the virtual legs in accordance with the spatial transformation, thereby forming a unitary representation of the $C_{4v}$ group. There exists a unitary transformation $U$ such that $\rho(g)$ can be brought into a block-diagonal form consisting of irreducible representations, i.e.,

$$\rho(g) = U \left( \bigoplus_{\Gamma,j} \rho_{\Gamma,j}(g) \right) U^{-1}, \tag{A.1}$$

where $\rho_{\Gamma,j}(g)$ denotes the $j$th copy of the matrix representation associated with the irreducible representation $\Gamma$.

As an example, we consider the blocks associated with the irreducible representation A1 of the $C_{4v}$ group, corresponding to the trivial representation. The basis vectors of these blocks span a subspace of $\mathbb{V}_A$ and correspond to the orthonormal basis $\{A_j\}$ introduced in the main text. Since A1 is a one-dimensional representation, and $\rho_{A1}(g) = 1$ for all $g \in C_{4v}$, it suffices to identify the set of eigenvectors of $\rho(g)$ with eigenvalue 1 for all $g \in C_{4v}$. To achieve this, we first construct the matrix representation $\rho(g)$ for each group element $g \in C_{4v}$. Starting with a group element $g_1$, we find the subspace $\mathbb{V}_A^{(1)} \subset \mathbb{V}_A$ consisting of eigenvectors of $\rho(g_1)$ with eigenvalue 1. We then proceed iteratively: within $\mathbb{V}_A^{(1)}$, we construct the matrix representation for a second group element $g_2 \in C_{4v}$ and extract the subspace $\mathbb{V}_A^{(2)} \subset \mathbb{V}_A^{(1)}$ consisting of its eigenvectors with eigenvalue 1. By repeating this process for all $g \in C_{4v}$, we eventually obtain the subspace of $\mathbb{V}_A^{(A1)} \subset \mathbb{V}_A$ that is invariant under the full group action (one may reduce the computational cost by only considering group elements that are the generators of the group). An orthonormal basis of this final subspace gives the desired set of $A_j$ tensors.

This procedure can easily be generalized to other one-dimensional irreducible representations of the $C_{4v}$ group. In those cases, instead of searching for eigenvectors with eigenvalue 1, one selects vectors corresponding to eigenvalue $\chi_\Gamma(g)$ for $g \in C_{4v}$, where $\chi_\Gamma(g)$ is the character of the corresponding one-dimensional irreducible representation $\Gamma$. Alternatively, as $C_{4v}$ is a finite group, one can explicitly construct the projector $P_\gamma \propto \sum_g \overline{\chi_\Gamma(g)}\rho(g)$ onto the one-dimensional irreducible representation $\Gamma$, and identify its eigenspace with eigenvalue 1.

## B  General form of local gauge transformations

In this section, we discuss the general form of local gauge transformations acting on the PEPS tensor. Starting from a general (translation invariant) gauge transform on all of the bonds of the network, it should be clear that preserving the spatially symmetric PEPS tensor $A$ and the constant bond tensor $D$ necessitates a transformation of the form in Eq. (2) that satisfies Eq. (3). Furthermore, because those tensors are chosen real, so should the gauge transform $G$ be a real invertible matrix. We now discuss the structure of $G$ in more detail.

In order to preserve the U(1) structure of the PEPS tensor, $G$ needs to be block diagonal with blocks $G_q$ acting on the subspaces $V_q$ of the virtual bonds. From Eq. (3) and the U(1) structure of the bond tensor $D$, we can further infer that

$$G_{-q}^T = G_q^{-1} \quad \text{for } q \neq 0, \quad \text{and} \quad G_{q=0}G_{q=0}^T = I. \tag{B.1}$$

Here, the subscript $q$ denotes the U(1) block of the matrix that corresponds to the U(1) charge $q$, and $I$ is the identity matrix. One can easily verify that all gauge transformation matrices form a continuous group. This implies that a sequence of gauge transformations can always be represented by a single transformation, allowing us to focus on a single gauge transformation in the following discussion.

Among all the gauge transformation matrices, there exists a class of gauge transformations that are unitary. A unitary gauge transformation $U$ satisfies the following condition:

$$U_{-q} = (U_q^T)^{-1} = U_q \quad \text{for } q \neq 0, \quad \text{and} \quad U_{q=0}U_{q=0}^T = I, \tag{B.2}$$

When applied individually, these local unitary gauge transformations do not alter the entanglement structure of the environments, and can be excluded from the discussions.

For an arbitrary gauge transformation $G$, we can always perform a polar decomposition $G = RU$, where $R$ and $U$ are both gauge transformation matrices, and, additionally, $R$ is symmetric and positive definite whereas $U$ is unitary. In combination with the U(1) structure, this implies that $G_q = R_q U_q$ and that the aforementioned properties hold for each of the diagonal blocks, i.e. $R_q$ is symmetric and positive definite, whereas $U_q$ is unitary (or orthogonal, as everything is real). In combination with the conditions in Eq. (B.1), this yields

$$R_{-q} = R_q^{-1}, \quad \text{for} \quad q > 0, \quad \text{and} \quad R_{q=0} = I, \tag{B.3}$$

$$U_{-q} = U_q, \quad \text{for} \quad q > 0. \tag{B.4}$$

The gauge transformation $G$ can be viewed as the composition of two successive transformations, where the second gauge transformation $U$ is unitary and can thus be omitted from the analysis. In the following, we only focus on positive definite gauge transformation matrices.

A positive-definite gauge transformation matrix $R$ can be written as $R = \exp(N)$, where $N$ is a symmetric matrix that satisfies

$$N_q = N_q^T = -N_{-q} \quad \text{for } q > 0, \quad \text{and} \quad N_{q=0} = 0, \tag{B.5}$$

where $N_q = \log(R_q)$. One can easily verify that $N$ anti-commutes with PEPS bond tensor $D$.

The set of symmetric matrices satisfying the condition (B.5) forms a linear space. A basis $\{N_\gamma\}$ of this space can always be chosen such that any matrix $N$ in the space can be expressed as $N = \sum_\gamma \tau_\gamma N_\gamma$, where $\tau_\gamma$ are real numbers.

## C  Hermiticity of $\sigma$ matrices

For each $N_\gamma$ in the basis $\{N_\gamma\}$, we can define a corresponding $\sigma_\gamma$ matrix. The $\sigma_\gamma$ matrix is given by

$$[\sigma_\gamma]_{ij} = \text{Tr}[A_i^\dagger \mathcal{L}_{N_\gamma}(A_j)], \tag{C.1}$$

where $\mathcal{L}_{N_\gamma}$ is the mapping from $A$ to $X$ defined in Eq. (4). Here, we take $A_i$ and $A_j$ as linear maps between composite linear spaces, and the multiplication and trace are defined in the sense of linear maps. Furthermore, we rely on the orthonormality of the basis choice $\{A_i\}$, such that $\text{Tr}[A_i^\dagger A_j] = \delta_{ij}$. Since $N_\gamma$ is Hermitian, it is straightforward to verify that $[\sigma_\gamma]_{ij} = [\sigma_\gamma]_{ji}^*$, which implies that $\sigma_\gamma$ is also a Hermitian matrix.

## D  Uniqueness of the gauge-fixed PEPS tensor and the gauge-fixing procedure

In this section, we discuss the uniqueness of the minimal canonical form (up to unitary transformations) and the gauge-fixing procedure in detail. Considering a gauge transform $G$ with the properties discussed in the previous sections, we have argued that we can omit from $G$ a unitary factor, as this does not affect the Frobenius norm of the gauge-transformed tensor $A$, and thus restrict to $G$ that are positive definite. This simplified the analysis in the main text.

In the current context, however, this restriction comes with the downside that the resulting set of gauge transforms is no longer a group, so that a single (positive definite) transform $G$ cannot be split as $G_1 G_2$ where both $G_1$ and $G_2$ are positive definite. Therefore, we now reintroduce the unitary gauge transforms and thus consider all real, U(1)-preserving, invertible matrices $G$ that satisfy Eq. (3). Now, we consider two tensors $A_0$ and $A_1$, represented respectively by $\vec{\alpha}_0$ and $\vec{\alpha}_1$, that are related by a gauge transform $G$. We will show that we can always construct a path of gauge-equivalent tensor $\vec{\alpha}(\mu)$ that connects $\vec{\alpha}_0 = \vec{\alpha}(0)$ and $\vec{\alpha}_1 = \vec{\alpha}(1)$, and along which the (squared) norm $\|\vec{\alpha}(\mu)\|^2$ is strictly convex, i.e. $\frac{d^2}{d\mu^2}[\vec{\alpha}(\mu)^T \vec{\alpha}(\mu)] > 0$. In particular, this means that, if $A_0$ had actually a minimal norm, as characterized by $\vec{\alpha}_0^T \sigma_\gamma \vec{\alpha}_0 = 0$ for all $\gamma$, then the norm monotonously increases along $\vec{\alpha}$, and the norm of $\vec{\alpha}_1$ is strictly larger. Even if other paths can be constructed along which the squared norm is not monotonous, this does not affect the result.

As before, we can decompose the gauge transform $G$ into its unitary and positive definite parts as $G = QR$. We will first apply the unitary factor $Q$ to $A_1$, obtaining a new tensor $A_1'$ with the same norm as $A_1$, and then connect $A_0$ to $A_1'$ instead. This does not affect the final conclusion that $\|A_1\|^2 = \|A_1'\|^2 > \|A_0\|^2$ if $A_0$ is a tensor with minimal norm. The importance of this step also comes from the fact that we have glossed over an important detail, namely that the set of gauge transformations might not be connected. Indeed, since we restrict ourselves to real numbers, the unitary subgroup of gauge transformations is actually a subgroup of O($D$), the group of orthogonal $D \times D$ matrices, which is known to have two components. Due to the U(1)-preserving condition and Eq. (3), this subgroup can further fragment into different components. In contrast, the space of positive definite matrices is connected to the identity matrix, and we will be able to connect $A_0$ to $A_1'$ along a continuous path.

Because a real symmetric positive definite matrix $R$ has (strictly) positive eigenvalues, it has a uniquely defined real logarithm $N^R = \log(R)$, where $N^R$ can be expanded in the basis of $N_\gamma$ as $N^R = \sum_\gamma \tau_\gamma^R \sigma_\gamma$. At the level of the vector representation, we now construct the path

$$\vec{\alpha}(\mu) = \exp(\mu \sigma^R)\vec{\alpha}_0 \tag{D.1}$$

with

$$\sigma^R = \sum_\gamma \tau_\gamma^R \sigma_\gamma. \tag{D.2}$$

This path has the desired boundary conditions $\vec{\alpha}(0) = \vec{\alpha}_0$ and $\vec{\alpha}(1) = \vec{\alpha}_1'$, the latter being the

vector representation of $A_1'$. Note that $\sigma^R$ is still a real symmetric matrix. We now find

$$\frac{\mathrm{d}^2}{\mathrm{d}\mu^2}\|A(\mu)\|^2 = \frac{\mathrm{d}}{\mathrm{d}\mu^2}\vec{\alpha}(\mu)^T\vec{\alpha}(\mu) = \vec{\alpha}(\mu)^T(\sigma^R)^2\vec{\alpha}(\mu) \geq 0. \tag{D.3}$$

Because $(\sigma^R)^2$ is the square of a real symmetric matrix, it is positive semidefinite, and the latter expression is nonnegative. We thus find that the first derivative

$$\frac{\mathrm{d}}{\mathrm{d}\mu}\|A(\mu)\|^2 = \frac{\mathrm{d}}{\mathrm{d}\mu}\vec{\alpha}(\mu)^T\vec{\alpha}(\mu) = \vec{\alpha}(\mu)^T\sigma^R\vec{\alpha}(\mu) \tag{D.4}$$

must be non-decreasing. In particular, if $\vec{\alpha}_0^T\sigma^R\vec{\alpha}_0 = \sum_\gamma \tau_\gamma^R\vec{\alpha}_0^T\sigma_\gamma\vec{\alpha}_0 = 0$, then, as $\mu$ increases, the derivative will become positive and the norm itself will grow.

The only exception would be the case where both the first and second derivative are zero throughout the whole path. In this case, the norm would remain constant, so that the minimal canonical form tensor would not be unique (up to unitaries). We cannot exclude this directly, because $\sigma^R$ can have zero eigenvalues and $(\sigma^R)^2$ is thus not necessarily not strictly positive definite. However, in that case it would follow that $\sigma^R\vec{\alpha}(\mu) = \vec{0}$, and thus $\vec{\alpha}(\mu)$ would be constant itself, i.e. $\vec{\alpha}(\mu) = \vec{\alpha}_0 = \vec{\alpha}_1'$. This can only happen if $A_1' = A_0$, or thus if $A_0$ and $A_1$ were related by a purely unitary gauge transformation.

This proof constitutes a particular example of the general result mentioned in Ref. [28], which is known as the Kemph-Ness theorem.

To find a tensor with minimal norm in practice, we can apply Newton's method. Whereas the general hessian may not be guaranteed to be positive definite everywhere, it becomes so under a parameterization, in which, at every step, we update the current state $\vec{\alpha}_n$ as

$$\vec{\alpha}_{n+1} = \exp(\sum_\gamma \tau_\gamma\sigma_\gamma)\vec{\alpha}_n \tag{D.5}$$

for some combination of $\tau_\gamma$. For such updates, the (squared) norm of the tensor would be updated as

$$\mathcal{N}_n(\{\tau_\gamma\}) = \vec{\alpha}_n^T\exp(2\sum_\gamma \tau_\gamma\sigma_\gamma)\vec{\alpha}_n. \tag{D.6}$$

We can compute the gradient of the (squared) norm at the point $\vec{\alpha}_n$ (i.e. $\tau_\gamma = 0$ for all $\gamma$)

$$g_\gamma[\vec{\alpha}_n] = 2\vec{\alpha}_n^T\sigma_\gamma\vec{\alpha}_n \tag{D.7}$$

as well as the hessian

$$h_{\gamma\gamma'}[\vec{\alpha}_n] = 2\vec{\alpha}_n^T[\sigma_\gamma\sigma_{\gamma'} + \sigma_{\gamma'}\sigma_\gamma]\vec{\alpha}_n = 4\vec{\alpha}_n^T\sigma_\gamma\sigma_{\gamma'}\vec{\alpha}_n. \tag{D.8}$$

With these ingredients, we can determine $\tau_\gamma$ from the linear system

$$g_\gamma[\vec{\alpha}_n] + \sum_{\gamma'} h_{\gamma\gamma'}[\vec{\alpha}_n]\tau_{\gamma'} = 0 \tag{D.9}$$

and compute $\vec{\alpha}_{n+1}$. In practice, it can be useful to add a regularization parameter $\mu \in (0,1]$ and instead set

$$\vec{\alpha}_{n+1} = \exp(\mu\sum_\gamma \tau_\gamma\sigma_\gamma)\vec{\alpha}_n. \tag{D.10}$$

By repeating this process until the norm of the gradient $g[\vec{\alpha}_n]$ goes below a certain threshold (e.g. $10^{-12}$), we can find the minimal point of $\mathcal{N}(\{\tau_\gamma\})$ which leads to the gauge-fixed PEPS tensor.

# E   Retraction and vector transport on the manifold of gauge-fixed PEPS

Given a normalized PEPS tensor on the gauge-fixed manifold, specified using the vector $\vec{\alpha}_0$ satisfying $\vec{\alpha}_0^T \vec{\alpha}_0 = 1$ and $\vec{\alpha}_0^T \sigma_\gamma \vec{\alpha}_0 = 0$, we now need to be able to move on the manifold in the direction of a tangent vector $\eta_0$. Let us denote the resulting trajectory as $\vec{\alpha}(s)$, where we thus want

$$\vec{\alpha}(s)^T \vec{\alpha}(s) = 1, \qquad\qquad \vec{\alpha}(s)^T \sigma_\gamma \vec{\alpha}(s) = 0, \quad \forall \gamma, \qquad (E.1)$$

to be satisfied for every $s$, with furthermore $\vec{\alpha}(0) = \vec{\alpha}_0$. If we furthermore denote $\vec{\eta}(s) = \frac{d\vec{\alpha}}{ds}(s)$, then by simply differentiating the conditions in Eq. (E.1), we obtain the conditions

$$\vec{\alpha}(s)^T \vec{\eta}(s) = 0, \qquad\qquad \vec{\alpha}(s)^T \sigma_\gamma \vec{\eta}(s) = 0, \quad \forall \gamma, \qquad (E.2)$$

which characterize $\eta(s)$ as a proper tangent vector, and thus also need to be satisfied by the initial direction $\eta(0) = \eta_0$.

In a Euclidean space, the trajectory would simply be chosen as $\alpha(s) = \alpha_0 + s\eta_0$ and $\eta(s) = \eta_0$. However, such a straight line is not lying on the manifold of gauge fixed (and normalized) PEPS, as it can be seen to violate the conditions at second order in $s$. Hence, we expect that along the trajectory, the direction $\vec{\eta}(s) = \frac{d\vec{\alpha}}{ds}(s)$ will need to be updated, in order to preserve the gauge fixing constraint. These changes should correspond to changing the norm or the gauge of the PEPS tensor, in order to compensate deviations away from the normalized and gauge fixed manifold. We thus expect that

$$\frac{d\vec{\eta}}{ds}(s) = c_0(s)\vec{\alpha}(s) + \sum_\gamma c_\gamma(s)\sigma_\gamma \vec{\alpha}(s) \qquad (E.3)$$

with $c_0(s)$ and $c_\gamma(s)$ scalar coefficients that need to be determined along the trajectory. Differentiating Eq. (E.1) a second time, or thus differentiating Eq. (E.2), we obtain

$$\vec{\alpha}(s)^T \frac{d\vec{\eta}}{ds}(s) = -\vec{\eta}(s)^T \vec{\eta}(s), \qquad \vec{\alpha}(s)^T \sigma_\gamma \frac{d\vec{\eta}}{ds}(s) = -\vec{\eta}(s)^T \sigma_\gamma \vec{\eta}(s), \quad \forall \gamma. \qquad (E.4)$$

Inserting the expansion in Eq. (E.3) in those equations leads to the linear system

$$c_0(s) = -\vec{\eta}(s)^T \vec{\eta}(s), \qquad \sum_{\gamma'} \vec{\alpha}(s)^T \sigma_\gamma \sigma_{\gamma'} \vec{\alpha}(s) c_{\gamma'}(s) = -\vec{\eta}(s)^T \sigma_\gamma \vec{\eta}(s), \quad \forall \gamma \qquad (E.5)$$

We furthermore find $\vec{\eta}(s)^T \frac{d\vec{\eta}}{ds}(s) = 0$, so that $c_0 = -\vec{\eta}(s)^T \vec{\eta}(s) = -\vec{\eta}_0^T \vec{\eta}_0$. The coefficient matrix $M_{\gamma,\gamma'} = \vec{\alpha}(s)^T \sigma_\gamma \sigma_{\gamma'} \vec{\alpha}(s)$ appearing in the linear system determining the $c_\gamma$ coefficients is positive definite, so that this system should always lead to a well-defined solution. A retraction can then be constructed by integrating the differential equations

$$\frac{d\vec{\alpha}}{ds}(s) = \vec{\eta}(s), \qquad\qquad \frac{d\vec{\eta}}{ds}(s) = -\vec{\eta}_0^T \vec{\eta}_0 \vec{\alpha}(s) + \sum_\gamma c_\gamma(s)\sigma_\gamma \vec{\alpha}(s). \qquad (E.6)$$

If there are other tangent vectors $\vec{\xi}_0$ at position $\vec{\alpha}_0$, we can transport them along this retraction as a vectors $\vec{\xi}(s)$ that evolve according to $\vec{\xi}(0) = \vec{\xi}_0$ and

$$\frac{d\vec{\xi}}{ds}(s) = x_0(s)\vec{\alpha}(s) + \sum_\gamma x_\gamma(s)\sigma_\gamma \vec{\alpha}(s) \qquad (E.7)$$

where the scalar coefficients $x_0(s)$ and $x_\gamma(s)$ are determined by differentiating the tangent space conditions

$$\vec{\alpha}(s)^T \vec{\xi}(s) = 0, \qquad\qquad \vec{\alpha}(s)^T \sigma_\gamma \vec{\xi}(s) = 0 \qquad\qquad \text{(E.8)}$$

which leads to the linear equations

$$x_0(s) = -\vec{\eta}(s)^T \vec{\xi}(s), \qquad \sum_{\gamma'} \vec{\alpha}(s)^T \sigma_\gamma \sigma_{\gamma'} \vec{\alpha}(s) x_{\gamma'}(s) = -\vec{\eta}(s)^T \sigma_\gamma \vec{\xi}(s). \qquad \text{(E.9)}$$

By virtue of Eq. (E.7) and Eq. (E.8), one can easily verify that this vector transport preserves inner products, i.e. for two different tangent vectors it holds that $\vec{\xi}_1(s)^T \vec{\xi}_2(s) = \vec{\xi}_1(0)^T \vec{\xi}_2(0)$.

In practice, the typical steps taking during optimization are not very big, so that these differential equations with a simple integration scheme. Even applying a single step of the Euler method can often be sufficient, and is what we do in practice. Any deviations from e.g. the normalization or gauge condition on $\vec{\alpha}(s)$ can be restored with an extra normalization and gauge fixing step at the end of the retraction.

# F   The perturbative construction of the initial PEPS tensor

In this section, we discuss the perturbative construction of the initial PEPS tensor for the case of unit filling [40]. As we are interested in the case of unit filling, we fix the U(1) charge of the auxiliary space to be $q = -1$. As mentioned in the main text, we fix the bond tensor $D$ as

$$q \leftarrow\!\!\!- D -\!\!\!\rightarrow -q = I_{d_q}, \qquad\qquad \text{(F.1)}$$

which means that the U(1) block corresponding to charge $q$ and $-q$ is an identity matrix of dimension $d_q$ ($d_q$ is the size of the U(1) block).

At the limit of $U \to \infty$, the ground state of the hamiltonian (1) is a Mott insulator, where particles are localized on the lattice sites. This state is a product state, and can be represented as a PEPS with virtual space $V_{q=0}$ with bond dimension 1. The PEPS tensor $A_0$ is given by

$$\qquad\qquad \text{(F.2)}$$

As $U$ starts to decrease, the ground state starts to delocalize. The first-order effect of particle hopping is the movement of particles between neighboring lattice sites, where vacancy sites and doubly occupied sites are adjacent to each other. To represent this in the PEPS ansatz, we need to increase the physical dimension of the PEPS tensor to $V_{q=0} \oplus V_{q=1} \oplus V_{q=2}$, and the virtual space to $V_{q=0} \oplus V_{q=1} \oplus V_{q=-1}$. The tensors $A_1^a$ and $A_1^b$ correspond to the vacancy sites and the double occupied sites, respectively. They are given by

$$\qquad\qquad \text{(F.3)}$$

$$\qquad\qquad \text{(F.4)}$$

where all other blocks in $A_1^a$ and $A_1^b$ are zero. The total PEPS tensor is then given by $A = A_0 + \alpha(A_1^a + A_1^b)$, where the plus sign between $A_1^a$ and $A_1^b$ follows from the first-order term in the perturbative construction [40], and $\alpha$ is the free parameter. To determine the value of $\alpha$, one only needs to compute the energies for different values of $\alpha$ and find the one that minimizes the energy. The energy results for the case of $U = 30$ are shown in Fig. 6. According to the results, we choose $\alpha = 0.21$ to generate the initial PEPS tensor.

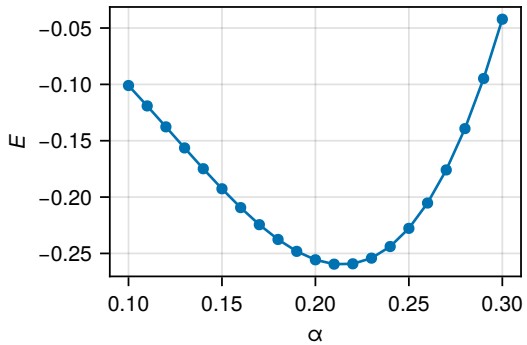

Figure 6: Energy results for the case of $U = 30$ as a function of $\alpha$.

We can already consider the effect of virtual gauge transformations on this simple PEPS tensor. In this case, there is only one remaining gauge degree of freedom, for which the $N$ matrix is given by

$$N_{q=1} = 1, \quad N_{q=-1} = -1, \quad N_{q=0} = 0. \tag{F.5}$$

Applying the gauge transformation $G = \exp(\tau N)$ to the PEPS tensor $A$, we obtain

$$A \to A_0 + \alpha \left( e^{-\tau} A_1^a + e^{\tau} A_1^b \right). \tag{F.6}$$

This expression indicates that the ratio between the coefficients of $A_1^a$ and $A_1^b$ does not affect the final PEPS state, and one can therefore freely choose this ratio. Fixing the ratio, or equivalently, setting a value for $\tau$, is equivalent to fixing the virtual gauge degree of freedom.

Recall that $A_1^a$ and $A_1^b$ correspond to sites with a vacancy and double occupancy, respectively, and in our case, the numbers of sites with vacancy and double occupancy should always be equal to each other. On the one hand, this explains why rescaling $A_1^a$ with a factor and $A_1^b$ with the inverse factor does not affect the state, but at the same time it suggest that the natural condition is for their coefficients to be equal, i.e. $\tau = 0$. One can easily verify that setting $\tau = 0$ minimizes the norm of the tensor $A$ with respect to $\tau$, so that choosing $\tau = 0$ is equivalent to imposing the MCF gauge-fixing condition. This indicates that the MCF gauge-fixing condition is a reasonable choice in many cases, as it seeks for a balance between tensor blocks with different U(1) charges.

Finally, as the virtual space for PEPS is set to $\mathcal{V}_{-1}^{\oplus 1} \oplus \mathcal{V}_0^{\oplus 2} \oplus \mathcal{V}_1^{\oplus 1}$ in our main text, we need to expand the bond dimension of the PEPS from 3 to 4. As this is only an initialization of the PEPS, it suffices to expand the PEPS virtual space by simply using a tensor $P$, which is a linear map from $\mathcal{V}_{-1}^{\oplus 1} \oplus \mathcal{V}_0^{\oplus 2} \oplus \mathcal{V}_1^{\oplus 1}$ to $\mathcal{V}_{-1}^{\oplus 1} \oplus \mathcal{V}_0^{\oplus 1} \oplus \mathcal{V}_1^{\oplus 1}$. The blocks of $P$ are given by

$$P_{q=1} = 0, \quad P_{q=-1} = 1, \quad P_{q=0} = \begin{pmatrix} 1/2 & 1/2 \end{pmatrix}. \tag{F.7}$$

The isometry tensor can be attached to the PEPS tensors as follows

$$\tag{F.8}$$

while the tensor $D$ is expanded according to Eq. (F.1).

# G   Contraction of the double-layer tensor network

In this section, we briefly discuss the contraction of the double-layer tensor network. Given a PEPS state $|\Psi\rangle$, we are interested in computing the expectation value of a local operator $\hat{O}$, which is given by $\langle\Psi|\hat{O}|\Psi\rangle / \langle\Psi|\Psi\rangle$. Both the numerator and the denominator of this expression can be represented as double-layer tensor networks.

As an example, consider the denominator $\langle\Psi|\Psi\rangle$. It can be represented as the double-layer tensor network shown in Fig. 7, where the tensors $T_D$ and $T_A$ are defined as

$$\tag{G.1}$$

Here, the open legs of $A$ and $A^\dagger$ in the same direction are fused into a single leg; the same applies to those of $D$ and $D^\dagger$. The double-layer tensor network for $\langle\Psi|\hat{O}|\Psi\rangle$ can be constructed analogously, with the only difference being that the operator $\hat{O}$ is inserted between the corresponding physical legs in the double-layer tensor network.

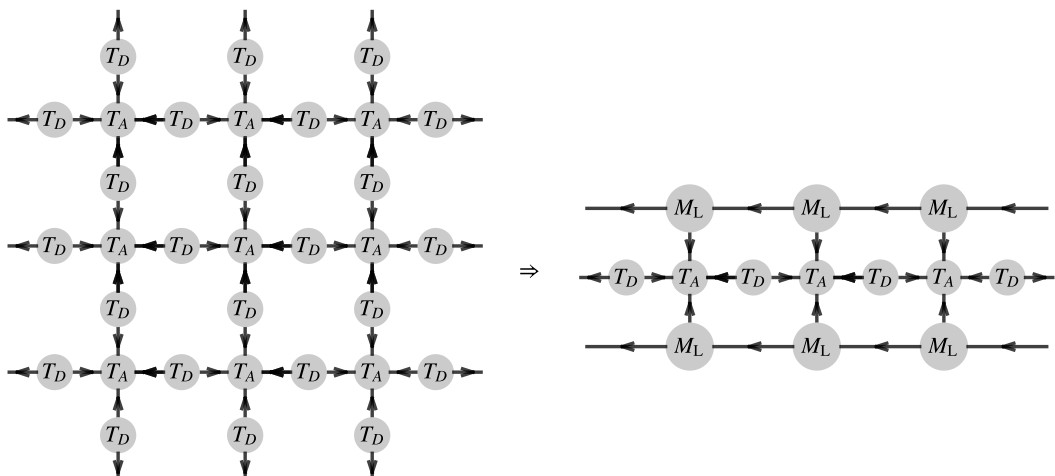

Figure 7: On the left, we show the double-layer tensor network representing $\langle\Psi|\Psi\rangle$. To contract this tensor network, we approximate the environments from the upper and lower directions by MPS, which is obtained by the VUMPS algorithm.

To contract the double-layer tensor network shown in Fig. 7, we employ the transfer matrix approach and approximate its leading eigenvector using an MPS. This MPS is obtained via the

VUMPS algorithm, which finds the fixed point of the transfer matrix within the manifold of MPSs with a fixed virtual space

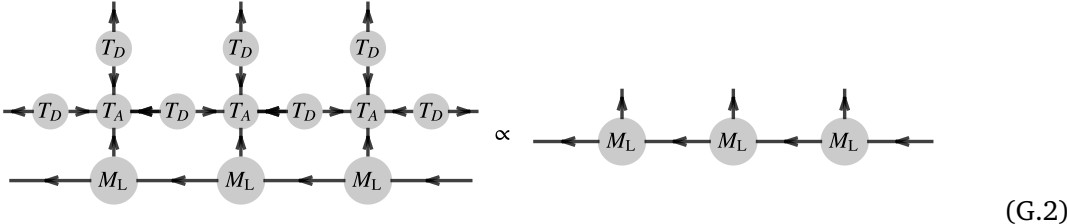

$$(G.2)$$

Due to the spatial symmetry of the PEPS tensors, we only need to compute the fixed-point MPS from the lower direction, as the environment from the upper direction can be directly obtained by flipping the tensor network vertically. With the environments approximated by MPSs, the contraction of the double-layer tensor network becomes straightforward, as shown in Fig. 7.

## H   The gauge-fixing condition with maximal fidelity

In this section, we discuss an alternative gauge-fixing condition, which is based on the transfer matrix approach that we use to contract the double-layer tensor network. For the sake of convenience, we denote the transfer matrix as $\mathcal{T}$, and its lower and upper eigenstates as $|\phi_{\mathrm{L}}\rangle$ and $|\phi_{\mathrm{U}}\rangle$. We take the convention that the transfer matrix $\mathcal{T}$ acts from top to bottom, i.e., $\mathcal{T}|\phi_{\mathrm{L}}\rangle \propto |\phi_{\mathrm{L}}\rangle$ and $\langle\phi_{\mathrm{U}}|\mathcal{T} \propto \langle\phi_{\mathrm{U}}|$. Due to the spatial symmetry of the PEPS tensors, the upper and lower eigenstates are related through the following equation

$$(H.1)$$

where we have used the condition on the bond tensor $D$ [c.f. Eq. (F.1)].

$$(H.2)$$

Recall the gauge transformation given by Eq. (2). This corresponds to a similarity transformation of the transfer matrix $\mathcal{T}$

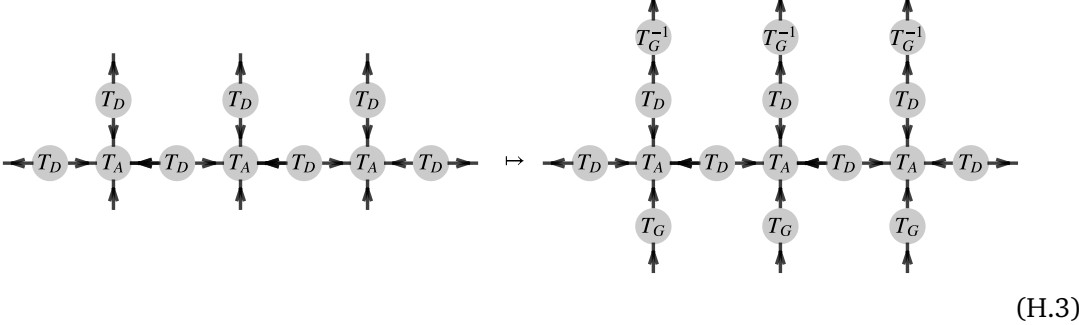

$$(H.3)$$

where the tensor $T_G$ is given by

$$(H.4)$$

We denote the similarity transformation as $\mathcal{T} \to \mathcal{G}^{-1}\mathcal{T}\mathcal{G}$, where $\mathcal{G}$ is composed of the tensor $T_G$. The similarity transformation alters the eigenstates accordingly, i.e., $\langle\phi_{\mathrm{U}}| \to \langle\phi_{\mathrm{U}}|\mathcal{G}$ and $|\phi_{\mathrm{L}}\rangle \to \mathcal{G}^{-1}|\phi_{\mathrm{L}}\rangle$.

Intuitively, the optimal gauge choice should be the one where the fidelity between the upper and lower boundary MPS, given by

$$F = \frac{|\langle\phi_{\mathrm{U}}|\phi_{\mathrm{L}}\rangle|}{\sqrt{\langle\phi_{\mathrm{U}}|\phi_{\mathrm{U}}\rangle}\sqrt{\langle\phi_{\mathrm{L}}|\phi_{\mathrm{L}}\rangle}}, \tag{H.5}$$

is maximized. One can easily verify that the overlap $\langle\phi_{\mathrm{U}}|\phi_{\mathrm{L}}\rangle$ is invariant under the gauge transformation, and $\langle\phi_{\mathrm{L}}|(\mathcal{G}^{-1})^{\dagger}\mathcal{G}^{-1}|\phi_{\mathrm{L}}\rangle = \langle\phi_{\mathrm{U}}|\mathcal{G}\mathcal{G}^{\dagger}|\phi_{\mathrm{U}}\rangle$ [c.f. Eq. (H.2)]. Therefore, maximizing the fidelity is equivalent to minimizing the norm of the gauge-transformed eigenstate $\langle\phi_{\mathrm{L}}|(\mathcal{G}^{-1})^{\dagger}\mathcal{G}^{-1}|\phi_{\mathrm{L}}\rangle$ with respect to gauge transformation $\mathcal{G}$.

In practice, the eigenstate $|\phi_{\mathrm{L}}\rangle$ is approximately represented by a boundary MPS. We compute the gradient of the norm function $\langle\phi_{\mathrm{L}}|(\mathcal{G}^{-1})^{\dagger}\mathcal{G}^{-1}|\phi_{\mathrm{L}}\rangle$ with respect to $\tau_{\gamma}$ at the point $\tau_{\gamma} = 0 \,(\forall\gamma)$, which is given by

$$g_{\gamma} = -2 \times \quad \text{(diagram)} \tag{H.6}$$

where the norm of the MPS is normalized to 1, and the tensors $\rho_{\mathrm{R}}$ and $\rho_{\mathrm{L}}$ are the left and right environment tensors of the MPS satisfying $\mathrm{Tr}(\rho_{\mathrm{L}}\rho_{\mathrm{R}}) = 1$. The tensor $T_{N_{\gamma}}$ is given by

$$\text{(diagram)} \tag{H.7}$$

which is obtained by taking derivative of $T_G$ with respect to $\tau_{\gamma}$.

The gauge-fixing condition can simply expressed as $g_{\gamma} = 0$, $\forall\gamma$. To reach this gauge-fixing condition, we can use a similar method as the one used in the MCF gauge-fixing procedure. For a given PEPS tensor, we compute the fixed-point MPS $|\phi_{\mathrm{L}}\rangle$ with VUMPS. Then, we compute the gradient (H.6) and the hessian of $\langle\phi_{\mathrm{L}}|(\mathcal{G}^{-1})^{\dagger}\mathcal{G}^{-1}|\phi_{\mathrm{L}}\rangle$ with respect to $\{\tau_{\gamma}\}$ at the point $\tau_{\gamma} = 0$, $\forall\gamma$. The hessian $h_{\gamma\lambda}$ is given by

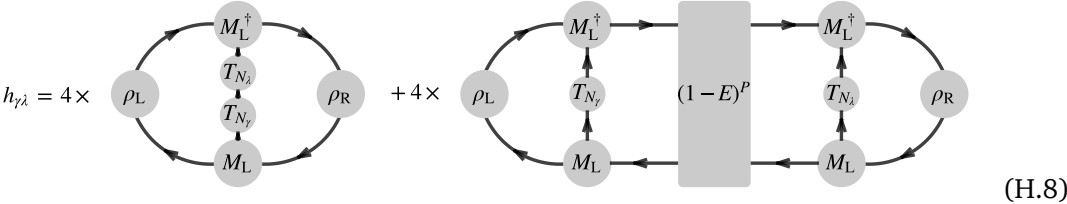

$$h_{\gamma\lambda} = 4\times \quad \text{(diagram)} \quad + 4\times \quad \text{(diagram)} \tag{H.8}$$

where $E$ represents the MPS transfer matrix, and the superscript $P$ represents the pseudoinverse. We take one step of the Newton's method, which gives $\Delta\tau_{\gamma} = -[h^{-1}g]_{\gamma}$, and then update the tensor as $\vec{\alpha} \to \exp(\sum_{\gamma}\Delta\tau_{\gamma}\sigma_{\gamma})\vec{\alpha}$. Using the updated PEPS tensor, we compute the fixed-point MPS $|\phi_{\mathrm{L}}\rangle$ again, and then repeat the above process until the norm of the gradient goes below a certain threshold.

There is a small subtlety here, in that the transformation behaviour $\langle\phi_{\mathrm{U}}| \to \langle\phi_{\mathrm{U}}|\mathcal{G}$ and similar for $|\phi_{\mathrm{L}}\rangle$ are only true for exact eigenvectors. Indeed, it is exactly this covariant transformation behavior that is destroyed by any practical algorithm that aims to approximate

these eigenvectors as MPS. However, this does not really affect the previous argumentation, as throughout the maximization of the fidelity, we will be recomputing the MPS approximations to $\langle\phi_{\mathrm{U}}|$ and $|\phi_{\mathrm{L}}\rangle$. This is also the reason why this gauge-fixing condition is much more expensive than the MCF gauge-fixing condition. Moreover, it is not clear how to perform the tangent space projection and the vector transport under this gauge-fixing condition. Nevertheless, we can compare the energies computed under this gauge-fixing condition with those obtained under the MCF gauge-fixing condition. More specifically, we take the PEPS tensors from each optimization step of the MCF-gauge-fixed PEPS optimization with the poorly behaved preconditioner [c.f. Fig. 5 in the main text], and then fix these tensors to the maximal-fidelity gauge and compute the energies again. As a comparison, we also perform the same procedure for the MCF-gauge-fixed PEPS optimization using the ordinary L-BFGS algorithm, where the optimization converges [c.f. Fig. 2 in the main text]. The results are shown in Fig. 8.

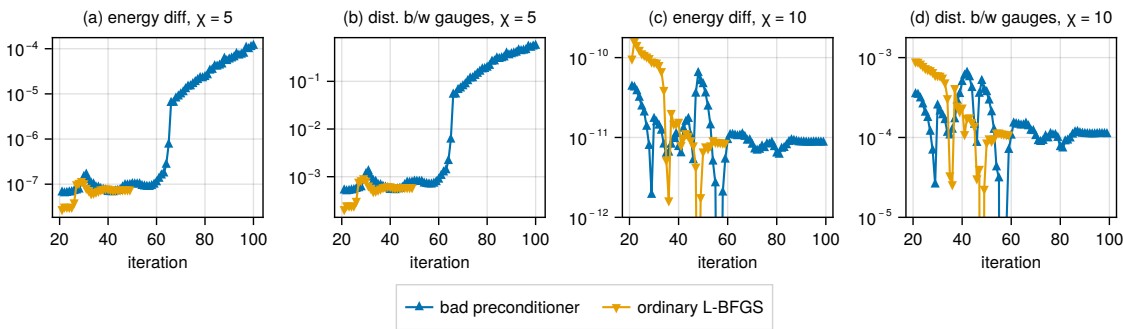

Figure 8: (a) Difference between the energies measured in the maximal-fidelity gauge and the MCF gauge, using PEPS tensors obtained from the MCF-gauge-fixed optimization with environment bond dimension $\chi = 5$ and a poorly conditioned preconditioner. (b) Distance between the two gauges as a function of the optimization step, using the same PEPS tensors as in (a). (c) Difference between the energies measured in the maximal-fidelity gauge and the MCF gauge, using PEPS tensors from the MCF-gauge-fixed optimization with environment bond dimension $\chi = 10$ and the standard L-BFGS algorithm. (d) Distance between the two gauges as a function of the optimization step, where the distance is quantified by the norm of $\tau$.

From Fig. 8(a), we observe that for the poorly preconditioned MCF-gauge-fixed PEPS optimization with $\chi = 5$, the difference between the energies measured in the maximal-fidelity gauge and the MCF gauge suddenly increases at a certain point during the optimization process. This indicates a large error is introduced in the energy measurement within the MCF gauge, under the assumption that the energy measured in the maximal-fidelity gauge is reliable. Such behavior explains the non-lower-bounded energy profile observed in the MCF-gauge-fixed PEPS optimization when using the poorly preconditioned L-BFGS algorithm. In contrast, the standard L-BFGS optimization, which converges properly, does not exhibit this issue.

To quantify the discrepancy between the two gauges, we measure the norm of $\tau$, which characterizes the gauge transformation required to convert the tensor from the MCF gauge to the maximal-fidelity gauge. We find that the behavior of this gauge distance exhibits the same pattern as the energy difference, confirming that the observed energy discrepancy arises solely from the gauge choice [see Fig. 8(b)].

We perform the same analysis for the case with a larger environment bond dimension, $\chi = 10$, where both optimization methods converge [see Fig. 8(c),(d)]. Although the distance between the two gauges remains only one order of magnitude smaller than in the $\chi = 5$ case, the corresponding difference in the measured energies is significantly reduced. This

suggests that increasing the environment bond dimension mitigates the impact of gauge choice on energy measurements, consistent with the observations shown in the main text [c.f. Fig. 5 in the main text].

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
