# Peer review of "Gauging the variational optimization of projected entangled-pair states"

_SciPost Physics_

## Round 1 · Referee Report · Anonymous (Referee 1) · 2025-11-18

Disclosure of Generative AI use

The referee discloses that the following generative AI tools have been used in the preparation of this report:

Used Apple Intelligence to correct typos and to improve grammar.

Strengths

1-Clear numerical evidence is provided for the claims made. In particular, unconstrained optimization drives tensors into pathological regions with artificially low energies. It is convincingly demonstrated that gauge-fixed optimization avoids this.

2-The explicit construction of gauge directions, the projection onto the tangent space, and the optimization of the manifold are technically sound.

3-The authors are very transparent about potential failures and do not conceal the instabilities within the MCF manifold when the search direction is distorted.

Weaknesses

1- Only one rather simple model is studied: a deep Mott insulator with a small D parameter and a tiny correlation length. While it would be interesting to consolidate the method by considering a more challenging model, the large numerical cost might make this very difficult.

2- No comparison with established optimization methods (imaginary-time / full update).

Report

The paper makes a significant contribution to the understanding and stabilization of gradient-based PEPS optimization. By demonstrating the role of non-unitary gauge transformations in destabilizing energy minimization, the authors shed light on the instabilities of unconstrained optimization.

Their proposed solution, constrained optimization using a minimal canonical form manifold with projected gradients, is simple and effective in the examined setting. Numerical evidence strongly supports the claim that gauge-aware optimization is more robust, producing physically meaningful tensors even when χ is small. The main limitations stem not from weaknesses in the method but from the narrowness of the test environment.

The study relies on a single model deep in a trivial phase and a symmetry structure that reduces gauge freedom to one dimension. Despite these limitations, the paper offers a significant and practical advance, providing a solid foundation for future work on gauge-aware PEPS optimization and identifying a clear direction for improving the reliability of tensor network methods.

Requested changes

1- If possible, include at least one test in a more entangled regime (larger correlation length, larger χ) to demonstrate robustness beyond the trivial case, e.g., pick a parameter near the Mott transition

2- If data is available, briefly compare to imaginary-time update methods to further demonstrated the importance of gauge-aware gradients.

Recommendation

Publish (easily meets expectations and criteria for this Journal; among top 50%)

---

## Round 1 · Referee Report · Anonymous (Referee 2) · 2025-12-7

Strengths

  • clear exposition
  • timely subject
  • implementable results that could possible be updated to a different systems

Weaknesses

  • closely related to previous work, see Ref 33 -results much geared towards specific system

Report

In this paper the authors consider the role of U(1) gauge transformations of PEPS descriptions. As shown in this work, see also previous work by these authors [Ref 33], gauge freedom can cause variations [and increasing different energies]. The authors reduce the U(1) case to a single class and present a gauge fixing strategy.

Overall I find this a timely and interesting work and would recommend publication. I have a few questions I like to pose before.

The authors restrict to the bosonic case and specifically restrict to the bose Hubbard model. In this case the U(1) tensor should transform [or be a combination of] irreps of the symmetry, e.g C_{4v}. How is their optimization affected when a ground state would have an order that is a combination of irreps. In particular from a symmetry point of view one could e.g. have an order parameter that has different parts that transform in the same irrep- especially if one considers a more difficult symmetry group than U(1). Does this not pose a problem for the optimization, also in view of the infinitesimal transformation above Eq.5 ?

In this regard I am also curious what the role of topology in the generalized construction would be. Indeed, in the free fermion case irrep order can related to invariants while in this construction one would could have an overall symmetry group that could allow a non-trivial invariant and perhaps even a ground state degeneracy in some more general cases. Note that overall symmetries can give simple PEPS states, e.g.
Nature Communications 16 (1), 284 (2025). As the gauge fixing is symmetry determined how would this be updated?

In fig 4 for \chi=15 the particle density fluctuates more than the others -of course the scale indicates the small number but still can the authors discuss the intuition here in a bit more detail?

Given the gradient-based approach I guess this could also be related to recent advances in ML techniques for optimization, which could be handy especially for more complicated symmetry groups and systems.

As a general question, could this be extended to the fermionic case in principle? I can see difficulties here; e.g. when one would TRS this would have transformation between different symmetry sectors.

As a minor comment the link to the appendix below Eq 6 is broken.

Recommendation

Publish (easily meets expectations and criteria for this Journal; among top 50%)

---

## Editorial Decision

awaiting_resubmission